# Engaging Parents of Lower-Socioeconomic Positions in Internet- and Mobile-Based Interventions for Youth Mental Health: A Qualitative Investigation

**DOI:** 10.3390/ijerph18179087

**Published:** 2021-08-28

**Authors:** Grace Broomfield, Catherine Wade, Marie B. H. Yap

**Affiliations:** 1Turner Institute for Brain and Mental Health, School of Psychological Sciences, Monash University, Clayton, VIC 3800, Australia; grace.broomfield@monash.edu; 2Parenting Research Centre, East Melbourne, VIC 3002, Australia; cwade@parentingrc.org.au; 3Faculty of Health Sciences, The University of Sydney, Sydney, NSW 2006, Australia; 4Melbourne School of Population and Global Health, University of Melbourne, Melbourne, VIC 3000, Australia

**Keywords:** qualitative, prevention, parenting, youth mental health, engagement, socioeconomic position, eHealth

## Abstract

Growing literature supports the use of internet- and mobile-based interventions (IMIs) targeting parenting behaviours to prevent child and adolescent mental health difficulties. However, parents of lower-socioeconomic positions (SEP) are underserved by these interventions. To avoid contributing to existing mental health inequalities, additional efforts are needed to understand the engagement needs of lower-SEP parents. This study qualitatively explored lower-SEP parents’ perspectives on how program features could facilitate their engagement in IMIs for youth mental health. We conducted semi-structured interviews with 16 lower-SEP parents of children aged 0–18 to identify important program features. Participants were mostly female (81.3%) and aged between 26 and 56 years. Transcriptions were analysed using inductive thematic analysis. Twenty-three modifiable program features important to lower-SEP parents’ engagement in IMIs were identified. These features aligned with one of three overarching themes explaining their importance to parents’ willingness to engage: (1) It will help my child; (2) I feel like I can do it; (3) It can easily fit into my life. The relative importance of program features varied based on parents’ specific social and economic challenges. These findings offer initial directions for program developers in optimising IMIs to overcome barriers to engagement for lower-SEP parents.

## 1. Introduction

Emotional and behavioural problems are a leading cause of disability in children and adolescents worldwide [1], making the prevention of these difficulties a major public health priority [2,3]. Parents of lower-socioeconomic positions (SEP) are underserved by parenting programs aimed at preventing and alleviating child and adolescent mental health problems, as these parents are less likely to enrol or engage in parenting programs compared to higher-SEP parents [3,4,5,6]. To address this inequity, additional efforts are needed to understand how programs can better meet the needs of socially and economically disadvantaged parents [3,7,8,9]. This paper aims to address this gap by exploring lower-SEP parents’ preferences for internet- and mobile-based parenting interventions (IMIs) for youth mental health. 

Growing literature supports the use of preventive interventions targeting parenting skills, parental self-efficacy and parent mental health to reduce the incidence and severity of child and adolescent mental health difficulties [10,11,12,13,14]. This evidence supports prevention and early-intervention across infancy [15], childhood [3] and adolescence [11,13]. As such, the present study has chosen a broad definition of child and adolescent mental health problems, with the term “youth” used to capture children and adolescents between the ages of 0–18. Despite the efficacy of preventive parenting interventions, their positive impact is often undermined by difficulties in engaging parents [16,17]. Studies indicate that enrolment rates of eligible parents vary greatly, from 10% to 90% [3,6,17,18,19] and up to a third of enrolled participants do not attend a single session [18]. This is a significant challenge for prevention efforts, as poor program reach undermines cost-effectiveness, and individual and community benefits [20].

One population uniquely underserved by parenting interventions aimed at preventing mental health difficulties are lower-SEP parents. SEP is one means of classifying people based on their relative access to social and economic resources [21], with individual or family factors often used as indicators of SEP or, alternatively, community-level contextual factors such as neighbourhood advantage [22]. Unfortunately, parents of lower-SEPs engage with face-to-face preventive parenting interventions at a lower rate than higher-SEP parents [3,4,5,6]. The under-engagement of lower-SEP parents is particularly concerning as children and adolescents within these families are at an elevated risk of mental health difficulties [23,24,25]. This poor engagement results in these families missing out on the potential benefits of prevention programs and can exacerbate existing mental health inequalities [26,27].

The internet has been identified as an alternative means of intervention delivery, which may expand the reach of preventive parenting programs and increase lower-SEP parents’ enrolment [28,29,30]. IMIs are predominately self-guided psychosocial interventions implemented through prescriptive online programs or mobile-based applications [31,32]. Meta-analyses indicate that preventive parenting IMIs can successfully reduce externalising and internalising difficulties in young people [33,34], with no significant difference in intervention effects found between IMIs and face-to-face parenting programs [35]. Examples of parenting IMIs that aim to prevent internalising difficulties include Partners in Parenting (PiP) [36,37,38], Cool Little Kids Online [39,40] and Parenting Resilient Kids (PaRK) [41,42] and IMIs that aim to prevent externalising difficulties include Triple P Online [43,44], *ez*PARENT [8,30,45] and ParentWorks [46,47].

IMIs offer several advantages over conventional approaches, which may support their uptake among lower-SEP families [48,49]. For example, lower-SEP parents often face logistical challenges with attending in-person programs, such as inflexible work schedules, access to transport, inconvenient location of services and lack of childcare [50,51]. Online delivery of programs means that parents can engage with these programs when and where it is most convenient for them whilst reducing many of the secondary costs associated with program completion, such as childcare expenses and lack of access to transport [52]. Psychological barriers also play an important role in low enrolment for in-person interventions, with the stigmatisation of mental illness [53,54] impacting lower-SEP parents’ willingness to engage [50,55]. Therefore, it has been suggested that digital delivery of programs may also promote engagement through normalising help-seeking and program use [56], as well as the additional anonymity afforded to parents online [57,58].

Despite the potential benefits of IMIs, lower-SEP parents remain underserved by these programs [59,60]. A contributing factor to this under-engagement is likely the underrepresentation of this population in samples used to develop and evaluate these programs [36,39,58,61,62]. This is problematic as lower-SEP parents experience unique barriers when engaging with IMIs, including more limited access to internet-enabled devices [63] and lower digital literacy [64]. As such, IMIs designed without consideration of how program features may interact with lower-SEP parents’ individual and environmental circumstances, are less likely to meet their needs [65].

Therefore, to ensure existing mental health inequalities are not exacerbated by the proliferation of IMIs that inadequately engage lower-SEP parents, programs need to be designed and tailored to better meet their needs. Traditional engagement and health behaviour change frameworks can provide some guidance (e.g., [65,66,67,68]), as can reviews of engagement-related factors (e.g., [16,17,69,70,71,72]). Responsible research and innovation principles [73] highlight the importance of obtaining user perspectives in IMI design and implementation to ensure the field is inclusive and responsive [74,75]. To date, qualitative studies investigating facilitators and barriers to engagement in preventive parenting programs have largely focussed on in-person interventions, program facilitator perspectives or have higher-SEP samples (see reviews [71,76]). One notable exception sought low-income parents’ perspectives on their use of the *ez*Parent program [77], providing insights into factors associated with lower-SEP parents’ ongoing engagement in an IMI. However, there remains a gap in the literature regarding program factors associated with lower-SEP parents’ initial engagement in preventive parenting IMIs for youth mental health. 

### Aims

The purpose of this study was, therefore, to qualitatively explore how modifiable program features can be used to increase lower-SEP parents’ engagement in preventive parenting IMIs for youth mental health difficulties. Research questions were: What modifiable program features do lower-SEP parents consider important to their engagement in IMIs for youth mental health? Why do they consider these features important to their engagement?

## 2. Materials and Methods

### 2.1. Study Design

This exploratory qualitative study consisted of semi-structured interviews with 16 lower-SEP parents. The interviews aimed to elicit the participants’ diverse perspectives on program features of preventive parenting IMIs for youth mental health and explore lower-SEP parents’ viewpoints on how program features could influence their engagement.

### 2.2. Participants

Lower-SEP parents or guardians across Victoria, Australia were recruited through flyers in schools, community organisations and youth mental health services, and digital flyers posted on social media pages of community organisations. Inclusion criteria were: (1) parent or guardian of a child aged 0 to 18 years; (2) 18 years or older; (3) fluent in English; (4) have an annual taxable family income of less than AUD 80,000, as per the Australian Government’s Family Tax Benefit A supplement payment eligibility criteria, which provides financial support to families based on their economic circumstances [78]. In total, 16 participants were recruited between March and August 2019. 

Participants were aged between 26 and 56 years (M = 39 years), who were parents or guardians of between 1 and 6 children (M = 2) aged between 1 month and 18 years (M = 8 years). Nine of the parents lived in metropolitan suburbs and seven lived in regional areas. Most parents were primarily engaged in either part-time employment (31.3%) or home duties (31.3%), and 56.3% were single parents. Most parents had not studied beyond vocational education (75.1%) and were living in a household with an annual taxable income of less than AUD 40,000 (56.3%). Additional sample demographics are outlined in Table 1. 

### 2.3. Procedure

Flyers were placed in local community centres, school newsletters and on parenting and community social media pages. Interested parents contacted the research team via an email or phone number listed on flyers and were provided additional information about the study. If interested, eligibility was assessed and parents were emailed an explanatory statement. The first author then contacted the participant to arrange a time and location for the interview. Locations were selected based on convenience for the participant and private rooms at local libraries or community organisations were generally chosen. Upon meeting at the agreed location, participants were reminded of the purpose of the study and that the interview would be audio recorded and transcribed verbatim. The interviews were conducted by the first author using a semi-structured interview guide and lasted between 35 and 95 min. Following the interview, all participants received an AUD 20 voucher for their participation. An initial sample of ten interviews were conducted with further interviews conducted until data saturation occurred. Data saturation was characterised as a further three interviews, following the initial ten, with no new themes emerging with analysis [79].

The study was approved by Monash University’s Human Research Ethics Committee (Project ID 17741). Informed consent was obtained from participants prior to commencing the interview via signed consent forms.

### 2.4. Measures

A semi-structured interview guide was developed to address the key domains of the research questions. Participants were asked to identify program features that may influence their engagement in a parent IMI for youth mental health and explain the importance of identified features. They were then asked to share their perspectives on a list of program features identified from prior research. The interview schedule (see Appendix A) was developed iteratively, with questions modified or added where new lines warranting investigation became apparent. 

### 2.5. Data Analysis

Audio recordings of the interviews were transcribed by the first author or a trained research assistant. Research assistants were trained by the first author to accurately, consistently and ethically transcribe the interview recordings verbatim. Participants were emailed a copy of their transcript and offered the opportunity to check their responses to address the internal and face validity of the data [80]. Data analysis was conducted by the first author with the aid of NVivo software [81]. 

Using a critical realist informed framework, transcripts were analysed using Braun and Clarke’s (2013) inductive approach to reflexive thematic analysis, allowing the data to drive theme generation [82]. A theme represented a patterned response regarding a program feature of importance or a factor influencing parental engagement. Following Braun and Clarke’s (2013) recommendations, each transcript was systematically read and analysed through the application of six phases: (1) familiarization with the data; (2) coding; (3) searching for themes; (4) reviewing identified themes; (5) defining and naming of themes; (6) finalising analysis. Thematic analysis concentrated on finding meaning across the data set, rather than exposing unique meanings within a single data item, and was recursive, involving movement back and forth between the stages [83]. Themes were selected based on prevalence but also whether the data provided a “rich description” [83] (p. 11). Themes and sub-themes were continually reviewed as the analysis progressed, with initial results reviewed by all authors. Only material that was irrelevant to the research questions or inaudible was excluded.

## 3. Results

Analysis led to the identification of 23 modifiable program features important to parents’ engagement in IMIs for preventing youth mental health difficulties. These 23 features were associated with seven program qualities and three overarching themes based on the belief activated in parents about a program due to the feature’s presence: (1) It will help my child; (2) I feel like I can do it; (3) It can easily fit into my life. Table 2 provides definitions for the three overarching themes. Figure 1 provides a visual representation of the relationships between these three overarching themes and their related sub-themes.

Each identified program feature was considered a facilitator of engagement due to its perceived ability to overcome or reduce one or more barriers associated with the parent’s participation in a parent IMI for youth mental health. Several of the program features were described as optimising more than one program quality, as can be seen in Figure 1. However, during analysis it was found that each program feature aligned with one particular overarching theme most strongly. As such, the following sections are divided into the three overarching themes, with each section detailing the associated program features, the barriers they overcome and parents’ reported preferences. Variability in parents’ reported preferences for each program feature is also described.

### 3.1. Theme 1: It Will Help My Child

Participants reported a general uncertainty regarding whether their family would benefit from a parent IMI for youth mental health. They reported limited awareness of the existence of such programs and felt they may only enrol in an IMI if their child or adolescent showed signs of a mental health difficulty. This suggests that limited mental health literacy, as well as limited awareness of the value of prevention, may be barriers to enrolment for lower-SEP parents in universal and selective prevention programs. 

Despite limited understanding of the benefits of such programs, parents identified eight program features that could be used to improve the perceived benefit of such programs. These sub-themes largely focussed on increasing the perceived effectiveness of the program or the perceived relevance of the program.

**User endorsements:** Most parents felt this was the most persuasive type of evidence supporting the efficacy of an intervention. Over half of participants were interested in hearing other users’ perspectives. These parents suggested that IMIs include reviews, ratings, testimonials or percentages of satisfied parents to demonstrate that other parents had found the program beneficial. This would increase their belief that such a program could also help them and their child.
*“[Have] comments or testimonials from people. I mean when I look into doing things I always look at reviews for stuff first, especially if it costs money.”*—Mother of one, 30 years old

**Professional endorsements:** Parents felt they may also be persuaded that a program could benefit their child if a professional endorsed the program. Parents were most interested in health professionals’ perspectives on a program, with maternal health nurses, paediatricians, general practitioners and psychologists cited as reliable sources. One parent indicated that associations with research institutes and universities may be viewed less favourably by parents with less education due to perceived elitism among such institutions.
*“I feel like there are a lot of people out there, not a lot but some, that won’t have participated in any sort of further education and may not have even finished high school and are kind of put off by the “you think you are better than me because you went to university.”*—Mother of one, 29 years old

**Empirical support:** Five parents indicated they would be more motivated to enrol in a program if they could see it had research supporting its development and implementation. Three of these parents suggested this could be communicated through a brief summary of the literature, with relevant research cited or linked.
*“If I am going to invest time into something and get advice on how to parent my child … I want to know beforehand and have faith beforehand that it is something based on good evidence.”*—Mother of two, 32 years old


**Clear benefits:** The communication of program benefits was linked to perceived effectiveness. The clarity with which benefits were communicated was important to parents, with two indicating research summaries may be less meaningful, or possibly confusing. Instead, they wanted a brief list of what they could expect for their child after finishing the program or, alternatively, key words indicating what previous parents had gained from the program.
*“[Benefits] need to be very clear, otherwise I won’t part with my precious money … have examples of how it can help you in your day-to-day life, what you can expect to get out of it.”*—Mother of one, 35 years old

**Program content:** Parents sought a program that had content relevant to their needs and experiences. Participants cared that a program would address issues related to their current circumstances, with this often connected to challenges relevant to their child’s developmental stage, such as school transitions, bullying or alcohol use. The presence of specific strategies to help manage behaviours was also important. If the initial program content focussed heavily on parenting, rather than child and adolescent challenges, parents felt that the benefits may be less apparent to them.
*“I would want explanations behind behaviours to better understand why my children might be doing the sorts of things they might be doing and then sort of strategies to help address things.”*—Mother of two, 32 years old

**Multiple users:** Several participants noted the benefits of allowing multiple users within a family to access the same program. They felt this would accommodate various caregiving arrangements and allow for better support for their child. Parents wanted to be able to include partners, ex-partners, grandparents, cousins, friends, siblings and children. Five parents wanted multiple parties to have access to the full program, suggesting multiple user profiles, whereas others suggested the inclusion of written summaries so that session content could be shared if needed.
*“I could see a lot of benefits from being able to have my husband do it, or my mum who looks after the kids, so that there is some consistency.”*—Mother of two, 32 years old

**Local resources:** Several parents wanted IMIs to be able to direct them to further services in their local area if they needed more information or wanted to speak to a professional. One participant suggested programs have a page where users can search their postcode and find additional community services. They felt this would enhance the sense that the program was tailored to their needs and could adequately support them regardless of their difficulties.
*“I feel like [people living regionally] have got less access to a lot of different resources … I find that a lot of GPs don’t really know what to do with you. So even a program that sort of helped you figure out who to turn to … or what your problem is and where to find help.”*—Mother of one, 35 years old

**Program navigation:** Finally, parents emphasised the importance of independent and flexible program navigation. Participants wanted to be able to exercise choice in what information or resources they had access to at any given time as this would allow them to optimise the relevance of the program and support their child better. It was suggested that this feature could include allowing users to select a subset of modules from a larger selection or allowing them to choose the order in which they complete sessions.
*“I know personally if I’ve got better concentration on the day I might try the more complex stuff and if I’m having an off day or a busy day where I’m trying to do everything at once I might try something that’s a bit simpler or less time consuming.”*—Mother of one, 30 years old

### 3.2. Theme 2: I Feel Like I Can Do It

Features within this theme facilitate the user’s confidence in completing the IMI. Parents reported factors such as mental health stigma, fear of judgement regarding their parenting, limited literacy and limited digital literacy as barriers to feeling capable and ready to engage with such a program. Despite these barriers, parents identified seven modifiable program features that could be used to increase their self-efficacy and subsequent willingness to engage. 

**Language complexity:** Eight parents spoke about the need for simple, easy-to-understand language, with five noting the specific importance of this for lower-SEP parents due to lower literacy and greater linguistic diversity within their communities. They advised that complex language may lead parents to doubt their ability to complete the program, reducing enrolment. To increase the approachability of the program for users with lower literacy levels, parents suggested limiting complex terms, replacing some written content with images, and providing pop-up definitions for complicated words or mental health terminology.
*“Well you could assume they would have less education, so nothing too complicated and nothing too scary … so that it’s, you know, welcoming, and the language is really simple.”*—Mother of one, 35 years old

**Inclusive imagery:** Parents wanted programs that used images that felt welcoming, inclusive and demonstrated the program was designed for families like them. This can be achieved by including images of families that lower-SEP parents connect with. One parent suggested using depictions of people with different skin tones, family compositions and not having any parents in suits or high-fashion clothing.
*“I think it would need to be a real family, and not a cookie cutter Pinterest family … really embrace that a happy family doesn’t need to be a perfect family. You know, there’s not a dad in a suit, and you know, he’s all vogue magazine and mum’s all Gucci bag.”*—Mother of one, 35 years old

**Translated content:** The value of having programs, or sections of programs, translated into different languages for linguistically diverse parents was conveyed by three participants. These participants recognised that a higher proportion of parents within their communities speak English as a second language. They therefore felt that having the program accessible in other languages could improve some parents’ confidence in completing such a program and increase the engagement of lower-SEP parents who speak multiple languages.
*“Some people can’t speak English so being able to access the information in their own language [is important] … they shouldn’t be excluded from information they may need.”*—Mother of one, 26 years old

**User interface:** Digital literacy was another barrier identified by parents, with several participants lacking confidence navigating websites and digital resources due to limited experience with computers. Participants suggested programs use a simple and easy-to-navigate user interface to reduce this barrier. Parents preferred a program that was bright and engaging, with navigation buttons clearly identifiable and not overly cluttered.
*“I think it needs to be clear and easy to see how to navigate through … If things look like text everywhere, then it doesn’t tend to look too appealing.”*—Mother of two, 32 years old

**Anonymous registration:** Parents identified that one of the benefits of IMIs is that they offer additional anonymity compared to face-to-face programs. They felt this may be particularly beneficial for parents living in communities with greater mental health stigma or those who fear their parenting may be judged by accessing such a program. One participant suggested that programs should offer parents the option to register with minimal personal details for those who are particularly concerned about privacy.
*“We’ve got a few [people in our community] that are just like ‘Why would you do it [seek mental health support], why do you want other people to know your business?’”*—Mother of one, 30 years old

**User-to-user support:** The desire to connect with other parents and share experiences was expressed by parents. The presence of a discussion forum or an associated social media page was viewed as an opportunity to normalise some of the parenting difficulties participants were experiencing in a supportive space and share ideas based on the program content. Whilst this was seen as a means of increasing parent confidence, one participant noted that the additional time associated with such a feature could be a burden for many lower-SEP parents and, therefore, felt any peer-to-peer contact should be optional.
*“I think it’s good if there are parents experiencing the same thing that they can all get on together and help one another because I know some mothers that are really struggling with their kids … and not having the support themselves.”*—Mother of three, 28 years old

**Professional support:** Despite parents’ desire for anonymity, access to a contact person or facilitator was advocated by five parents as a means of improving the approachability of IMIs. It was noted that a personal interaction when a parent first enquires about, or registers for, a program could make the program seem less daunting. Additionally, the option to have regular check-ins or to reach out to someone appealed to many parents as this would allow them to ask questions and feel supported throughout the program, particularly if difficulties arose. Parents’ preferred means of contact with a facilitator varied, with suggestions including phone calls, emails, video-conferences and live-chat messaging.
*“Sometimes you can get questions or sometimes you might have a specific example from your life that is a bit outside the norm so even if … at certain points you touch base with them, but not all the way through, I don’t think.”*—Mother of one, 35 years old

### 3.3. Theme 3: It Can Easily Fit into My Life

This theme revolved around participants’ desire for a program that was convenient, affordable and flexible so that it would easily fit into their life. Parents described the competing demands they experience daily and the need to prioritise their limited resources, indicating that a prevention program would usually not take precedence over more immediate concerns. It was felt that by optimising the features listed in this section, barriers associated with limited time, finances and internet access could be reduced. Eight modifiable program features were identified that may facilitate parents’ sense that an IMI for youth mental health can easily fit into their life. 

**Length of modules and number of modules:** Reducing the time required to engage with an IMI was important to parents as this would allow them to more easily fit the program into their busy lives. Three parents outlined that they would like the overall program to be as short as possible, with fewer modules, whereas the remaining parents placed less importance on the overall program length and more on the length of individual sessions or modules. There was little consensus on the ideal length or number of modules, with the former ranging from 15 min to 60 min and the latter ranging from a single-session program to 20 modules. Whilst the desire for sessions to be “short” and “quick” was salient, three parents noted they did not want the quality of the content to be sacrificed for the sake of convenience. Therefore, these parents felt it was important to balance program and session length, ensuring that sufficient skill and knowledge was still attainable.
*“My day includes commitments that are scattered across the day with patches of dead time … so the shorter the session is, the smaller the amount of dead time you need to put it in. If you could put it in a quarter of an hour that’s, you know, while you wait for your McDonalds to be ready.”*—Father of three, 48 years old

**Downloadable content:** Limited internet access was a barrier to engagement for four lower-SEP parents due to their homes not having reliable internet coverage or the costs associated with internet use, particularly phone data. One suggested solution for overcoming this barrier was to have content that could be downloaded. Parents indicating that they could then utilise free Wi-Fi at locations such as libraries or fast food outlets to download the necessary content and then interact with it at a more convenient time.
*“You’d have to work out your data before you start the program, if it’s going to be enough, or are you going to have to wait until, you know, your data’s back up to speed or something.”*—Father of three, 38 years old


**Ongoing availability:** Continuous access to the program was an important consideration to five parents’ enrolment. This feature was related to the ability to revisit content after the program was completed and also the availability of the IMI at any time through the day. Parents felt this would help them fit the modules into the most convenient moments for them, rather than having to log in at predetermined, and potentially inconvenient, times.
*“Children aren’t predictable so you can’t really predict when you are going to have time to do it.”*—Mother of two, 31 years old

**Program platform:** Over half of participants felt the platform or device through which they access the program was important to the accessibility of the program. Eight parents indicated a strong preference for a mobile- or tablet-based application, rather than a website, with two participants not owning a desktop or laptop computer. They noted that an application would allow them to download content as needed and also access the program at any time. Four parents also indicated a desire for the program to be accessible across devices so they could use a different device based on their location.
*“Ideally you could jump between different [devices] … You’ve got the website but you’ve also got the app and both of them access all your information.”*—Mother of one, 26 years old

**Program cost:** Participants described how limited financial resources had been a barrier to them accessing parenting or youth mental health services previously. This was mostly discussed in the context of secondary costs associated with completing a parenting program such as obtaining childcare, transportation costs, or in the case of IMIs, data usage. Several parents also seemed to be unaware of low-cost or free programs or services. Participants explained how despite wanting to engage in programs or services, they felt they lacked the financial flexibility to do so, with two parents expressing concerns that doing so would have resulted in difficulty meeting basic needs such as food and shelter. This financial struggle left parents feeling marginalised and isolated.
*“Financial constraints can be very conflicting, especially with people who are struggling to make ends meet and they need that information so they can help their children. We don’t want to feel like … bad parents.”*—Mother of one, 26 years old

**Payment options:** Despite costs often being a significant barrier for these parents, they were hopeful that the virtual nature of IMIs could facilitate more affordable options than face-to-face programs. Options for reducing the impact of limited financial resources on enrolment included alternative payment arrangements such as concession rates, trial periods, payment plans and pay-by-module options.
*“I think cost would be a big factor for us in the low-income bracket … I think that is what stops people from getting services they need so just be flexible.”*—Mother of two, 39 years old

**Content format:** A range of content formats were viewed as acceptable by parents, including videos, audio, text, graphics and cartoon sequences. Preferences often differed based on literacy levels, preferred learning styles and whether parents planned on completing the program whilst doing other tasks. As such, providing users with choice in the format of content may facilitate enrolment.
*“Audio would be fantastic because … in the car, driving from one place to another … That said, something that I’m clicking away on I’m probably engaging with more than something that’s pure audio so interactive would also be good to cement it in.”*—Father of two, 46 years old

### 3.4. Diversity in Preferences 

Across the sample, there was variation regarding the perceived importance of program features, based on parents’ individual circumstances. These preferences were informed by a parent’s experience of factors commonly associated with socioeconomic disadvantage such as limited education or income, as well as factors such as geographic location, relationship status and gender. This heterogeneity is described below. 

Differences in education and income levels across the sample appeared to impact how parents viewed certain program qualities. For example, parents in the lowest income brackets reported features associated with affordability, such as program cost, payment options and downloadable content, as central to their engagement. In contrast, parents with no, or limited, post-secondary education reported that features facilitating convenience would be particularly important to their willingness to enrol in an IMI for youth mental health. Education also appeared to be associated with digital literacy, with parents with less education reporting lower confidence in using the internet. These parents subsequently reported that they would be easily deterred if a program was not simple and easily navigated. Responses also indicated that parents with less education were particularly aware of the social stigma associated with using a parenting program and consequently desired features optimising approachability. These parents also suggested that links to higher education institutes may undermine their willingness to engage, whereas parents in the sample with greater education were motivated by university-conducted research. 

Geographic factors also impacted parents’ reported preference. Parents who lived in regional areas (*n* = 7) reported more limited access to face-to-face mental health services, and therefore sought a program which could meet a greater variety of needs, compared to those living in metropolitan areas. Consequently, regional parents were particularly interested in having regular access to a professional support person to whom they could direct questions or concerns. These parents also expressed a strong desire for programs to facilitate contact with other services if needed, for example, including lists of services and resources specific to their local area. Internet access was a barrier to engagement for several regional parents and, therefore, downloadable content or application-based programs may also support their engagement. Additionally, several regional parents reported that they were more likely to enrol if the advertising material demonstrated that the IMI was designed for, or tailored to, parents in their community, whereas this was not raised as a concern for parents living in metropolitan areas. 

Single parents in this study (*n* = 9) appeared to be more focussed on the convenience of the program than their partnered peers. The desire to optimise this characteristic was driven by the sense that they had more limited available time to complete such programs. They were focussed on the importance of brief modules and accessibility on phones and tablets so that they could complete sessions whilst out of the home doing other activities. 

Gender also appeared to influence the perceived importance of different program features. Lower-SEP fathers reported a desire for programs to focus on building knowledge, rather than developing skills, whereas mothers appeared to want a more comprehensive program that supported the development of skills to manage behaviour. Fathers also reported a preference for shorter, more concise programs and appeared less interested in social or interactive features such as facilitators or user-to-user support. In contrast, mothers appeared to express a greater interest in longer, broader programs that included lots of practical strategies and a facilitator who could address specific questions.

## 4. Discussion

Lower-SEP parents are underserved by preventive parenting IMIs [3,59,60], which is concerning as lower-SEP young people are at an elevated risk of mental health difficulties [23,24,25]. This study provides insights into how specific features of IMIs can be adapted to increase reach among this population and why these features may be important to lower-SEP parents. Similarities and differences with previous literature are outlined below, as well as key implications for program design and delivery.

In their efforts to describe and explain program features important to their enrolment, parents often spoke about the qualities they were seeking in an IMI, as well as the reasons they were seeking such qualities, with three overarching themes capturing the mechanisms through which program features may facilitate parental enrolment. These three themes align with several concepts described in existing health behaviour change frameworks, e.g., [66,68] and parental engagement models, e.g., [65,67]. However, the present study is most consistent with Randolph and colleagues’ (2009) framework for engaging parents in prevention, likely due to the prevention focus of this model. The overarching themes presented in this study and Randolph and colleagues’ (2009) framework both include constructs related to self-efficacy, family-related practical barriers and expectations of benefit for one’s child. Randolph and colleagues’ (2009) framework also includes discrete constructs of “perceived child susceptibility and severity” and “value new behaviour”, which were not included in our model. This difference may lie in the present study choosing to only address factors directly associated with program design. 

Whilst similar to previous frameworks, the tripartite model from the current study adds to the existing literature by highlighting how different program features align with different motivations for enrolling, particularly for lower-SEP parents. This differs to previous models, where the focus is largely on the ways in which parents vary in their capacity or readiness to engage (e.g., beliefs, demographics, norms; [66,68]) Instead, the present findings highlight the ways in which programs vary in their capacity to engage lower-SEP parents. We hope this offers a shift where, rather than placing the responsibility of engagement on these parents, who have fewer social and economic resources at their disposal than their peers, the responsibility is placed on our field to produce preventive parenting IMIs that take gradual steps towards addressing the dearth of appropriate programs for lower-SEP families [60]. 

The program features identified as important to lower-SEP parents’ enrolment in IMIs differ to those previously deemed relevant to other types of parenting programs, e.g., [16,17,69,70,71,72]. This is largely due to the digital nature of IMIs and the specific needs of this population because most previous research has focussed on in-person interventions and higher-SEP groups. For example, program location and childcare availability have been associated with the perceived convenience of face-to-face interventions [18,71,76,84]; however, these in-person features were not relevant in the current examination of IMI features. Instead, participants in this study described length of modules, number of modules, ongoing availability, content format and program platform as central to optimising convenience due to the digital nature of the intervention. This indicates that engagement enhancement strategies previously identified as important for lower-SEP parents’ engagement in face-to-face programs may not translate to IMIs. 

Some of the features identified in the present study have not been considered important in the broader IMI literature, whereas others are already prioritised in program design. One feature not currently widely accessible in IMIs is flexibility in content format (e.g., video, audio, text). A recent systematic review of technology-assisted parenting programs [60] found that none of the 14 IMIs identified provided users the choice to complete the whole program using their preferred format. Examples of features already prioritised include flexibility in program platform and program navigation options. For example, PiP [37,38] and ParentWorks [46] include navigation options, whereby parents can tailor their choice of modules based on their preferences, whereas parents accessing ParentWorks [46] or PaRK [41,42] can access these interventions across a range of platforms. However, despite the presence of these desired features, these programs still reported lower enrolment rates among lower-SEP parents compared to higher-SEP parents. This suggests additional research is needed to investigate the optimal integration of desired program features. For example, based on evidence associated with face-to-face programs [57,69], promotional material may need to better highlight the presence of appealing features to motivate enrolment. 

The variability in parents’ preferences across the present sample highlights an additional challenge in addressing the under-engagement of lower-SEP parents. The within-group variability documented in the present study, as well as within previous samples [85], suggests that program features likely to engage certain lower-SEP parents may be unsuccessful, and potentially obstructive, for other segments of the lower-SEP population. For example, higher-educated lower-SEP parents indicated that research summaries and empirical support would facilitate their engagement, whereas lower-educated parents in the sample suggested it may undermine theirs. This highlights that lower-SEP parents are a heterogeneous group, with different program preferences based on their specific social and economic challenges. This variability creates challenges for creating universal and selective prevention IMIs that effectively appeal to a large proportion of lower-SEP parents. One avenue for investigation is the increased customisability that IMIs offer, as expanding customisability would allow parents to tailor programs to their specific needs and preferences [86]. Additionally, multi-level prevention approaches, such as those implemented by PiP [36,37,58] and Triple P [43,44,61], offer IMIs of varying intensities and formats, broadening the available prevention options for parents. Approaches such as these could also increase the reach of prevention programs and better serve the varying needs that exist within lower-SEP communities.

### 4.1. Strengths and Limitations

A strength of this study was its success in reaching a proportion of the population often not represented in parenting research. All parents in this sample had a household income below the national average [87], and overall the sample had lower rates of tertiary education and metropolitan postcodes than the national average [88,89]. Although a relatively broad household income range was used as an inclusion criterion, this allowed for a greater range of social and economic challenges to be captured during the interviews. The subsequent transcripts provided perspectives often overlooked in the design and implementation of IMIs for youth mental health [36,39,58,61,62].

A methodological limitation of this study is that some of the data obtained during the interviews was not relevant to the present study. Throughout the data collection process, efforts were made to ensure participants were accurately conceptualising IMIs for youth mental health; however, certain responses suggested that some parents lacked a clear understanding of these programs or became confused at times. For instance, one parent discussed the physical health benefits of programs for children. Where possible, parents were corrected and data unrelated to IMIs was not analysed, but some data may have been erroneously included in analyses if it was unclear that the participant was unsure. 

Finally, this study focussed on how programs can be optimised to facilitate the engagement of lower-SEP parents. Therefore, the research questions, interview schedule and focus of analysis fostered the identification of themes related to modifiable program features. This allowed for the development of findings that will support program developers to tailor programs based on lower-SEP parents’ preferences. However, other engagement-related factors for lower-SEP parents (e.g., past program experience, mental health stigma, help-seeking norms) would have been overlooked by our narrow focus on modifiable program features. As such, further research is needed to provide a more exhaustive understanding of the factors impacting lower-SEP parents’ engagement in web-based parenting programs for youth mental health to comprehensively address under-engagement with this population.

### 4.2. Implications

Findings from this study offer directions for developers of parent IMIs to prioritise the inclusion and optimisation of program features that may overcome barriers to engagement for lower-SEP parents. These parents consider whether a program will benefit their child, whether they feel capable of completing it and whether it fits into their daily life. Therefore, IMI-program developers may be able to increase lower-SEP parents’ engagement in their programs by taking these themes into consideration and including associated program features [65]. It may also be important to clearly highlight the inclusion of such features in promotional materials to prompt initial engagement [57,69]. Additional considerations include increasing the customisability of IMIs to allow parents to tailor program features to best suit their needs [86] and the development of more multi-level approaches (e.g., [14,58]) to increase reach. Without these developments in the field, lower-SEP families will likely continue to be underserved by IMIs for youth mental health. 

Findings also suggest several future research directions. First, this study highlighted the importance of qualitative research in the early stages of program development, particularly for underserved populations such as lower-SEP parents. Whilst this study focussed on program features, it would be useful for future research to further investigate the interactions among individual, environmental, program and provider factors that are important to parental engagement [65], as well as family functioning [67], to provide a more comprehensive picture of engagement behaviour for lower-SEP parents. 

Finally, empirical evidence is needed to support the use of the identified program features. One necessary step is randomised controlled trials of programs that incorporate relevant features. However, to allow program developers to effectively optimise programs based on findings from this study, it would be useful to first obtain stakeholder views to determine the feasibility of identified features and then demarcate which of the identified program features have the greatest influence on the engagement rates of lower-SEP parents. One suggested method for obtaining such data from difficult-to-reach populations is discrete choice experiments [3]. This consumer preference-based method has been previously used to determine how treatment format preferences influenced parent participation in a parenting program [90]. Parent preference data can also determine the extent to which other sociodemographic features impact lower-SEP parents’ preferences for specific program features.

## 5. Conclusions

This article aimed to explore the perspectives of lower-SEP parents regarding program features that may facilitate their engagement in IMIs to prevent youth mental health difficulties. Parents considered 23 modifiable program features when choosing whether or not to enrol in an IMI. These findings offer insights for program developers to tailor programs accordingly. However, preferences for program features varied across the sample, suggesting that a “one-size-fits-all” approach may not be appropriate for optimising reach within lower-SEP communities. Customisable universal prevention programs and multi-level prevention approaches therefore warrant further investigation, as well as research to determine the relative impact of identified program features on lower-SEP parents’ engagement. 

## Figures and Tables

**Figure 1 ijerph-18-09087-f001:**
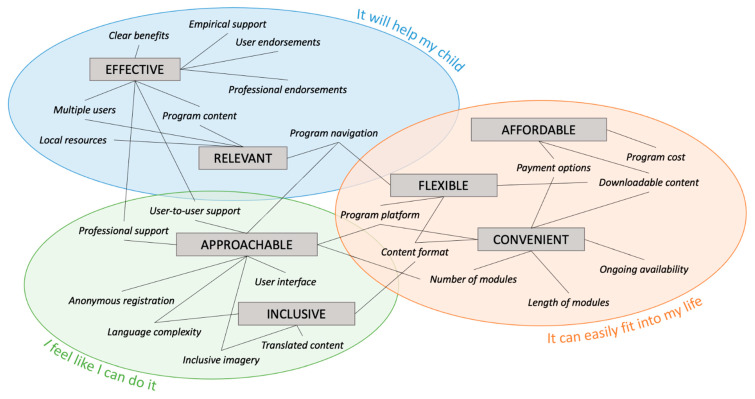
Thematic map illustrating the tripartite classification of themes and the relationships between themes and sub-themes.

**Table 1 ijerph-18-09087-t001:** Sociodemographic characteristics of interview participants (*N* = 16).

Variables	Participant Characteristics
Parent age in years; Mean (range)	39 (26–56)
Child age in years; Mean (range)	8 (0.08–18)
Number of children; Mean (range)	2 (1–6)
Relationship to child; *n* (%)	
Mother	12 (75.0)
Father	3 (18.8)
Guardian	1 (6.3)
Relationship status; *n* (%)	
Single	6 (37.5)
Married	4 (25.0)
De facto	3 (18.8)
Divorced	2 (12.5)
Widowed	1 (6.3)
Employment status; *n* (%)	
Full-time	3 (18.8)
Part-time	5 (31.3)
Contract	1 (6.3)
Home duties	5 (31.3)
Student	1 (6.3)
Unemployed	1 (6.3)
Highest education qualification; *n* (%)	
Primary school	1 (6.3)
Secondary school	5 (31.3)
Vocational education	6 (37.5)
Undergraduate degree	3 (18.8)
Postgraduate degree	1 (6.3)
Annual taxable household income (AUD); *n* (%)	
<AUD 20,000	3 (18.8)
AUD 20,000-AUD 39,999	6 (37.5)
AUD 40,000-AUD 59,999	3 (18.8)
AUD 60,000-AUD 79,999	4 (25.0)
Postcode; *n* (%)	
Metropolitan	9 (56.3)
Regional	7 (43.7)

**Table 2 ijerph-18-09087-t002:** Definitions of overarching themes from the parent interviews.

Theme	Definition
It will help my child	Program features facilitate the user’s belief that the program will lead to positive change for their child.
I feel like I can do it	Program features facilitate the user’s confidence that they are able to adequately engage with the program.
It can easily fit into my life	Program features facilitate ease of access, with the user feeling that the program does not interfere with existing demands.

## Data Availability

There are ethical restrictions on sharing our data. Data contains potentially identifying and sensitive patient information. According to the ethics approval obtained from Monash University’s Human Research Ethics Committee only the authors of this study and approved members of the research team can have access to the data.

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
