# Peer review of "Engaging Parents of Lower-Socioeconomic Positions in Internet- and Mobile-Based Interventions for Youth Mental Health: A Qualitative Investigation"

_ijerph, 2021, doi:10.3390/ijerph18179087_

Round 1

Reviewer 1 Report

This is a well-written research paper. However, it raises some doubts about the contents of the table that describe the themes and subthemes derived from the qualitative research study. For example, program content, multiple user, number of modules, is considered a different feature from inclusive imagery, user to user support, and downloadable content. I wonder if it is necessary to present a table. Also, I wonder if the contents of 3.4 from line 450 to line 496 are necessary. 

Reviewer 2 Report

I commend the authors' efforts on the work that has gone into this article which aimed to explore the perspectives of lower-SEP parents regarding program features that would facilitate their engagement in IMIs to prevent youth mental health difficulties. They make an important contribution to the literature on strategies to reach disadvantaged families. I have the following comments for consideration: 

Introduction

Line 46: '...difficulties in engaging...'

Line 64: Use the abbreviated 'IMIs' as it has already been written out in full in an earlier section

Lines 57 and 59: Repetition of 'under-engagement' in two consecutive sentences makes the reading tedious

Line 71: The sentence that begins with 'Whereas...' seems incomplete

Line 77: The challenge should be presented as '...lack of access to transport..'

Line 78: Suggestion to change it to 'Online delivery of programs...'

Line 83: Consider revising the statement as 'online' appears twice

Line 95: Suggestion to use 'adequately engage' instead of 'are adequately engaging'

Line 98: The use of 'However' at the beginning of the statement is misplaced

Materials and methods

At the beginning of this section, it would be worthwhile to explicitly provide  information on the study design

The title of the study makes reference to youth, but the recruitment covers children between the ages of 0 - 18 years. Some definitions suggest that youth are those aged 15-24 years. Consider revising the title to include 'children'

Lines 125-128: Move this from the section of participants as it is more appropriate to the subsection on 'Procedure'

Line 138: Under this subsection, the text needs to be re-organized. For instance, information on ethical considerations should be in one paragraph

Line 145: Who conducted the interviews?

Line 160: Who trained the research assistant? And what aspects did the training cover?

Lines 162 and 167: These two sentences should be combined 

Line 187: This sentence is not clear. 

Line 201: Rather than indicating that a program feature was linked to more than one program quality, it will be more meaningful to indicate specific aspects of quality 

Results

Line 359: How many parents?

Line 374: I suggest avoiding the use of positional statements such as 'below' as text position may change 

Discussion

The authors should refrain from reporting results under the discussion section again
